# Constitutive, Basal, and β-Alanine-Mediated Activation of the Human Mas-Related G Protein-Coupled Receptor D Induces Release of the Inflammatory Cytokine IL-6 and Is Dependent on NF-κB Signaling

**DOI:** 10.3390/ijms222413254

**Published:** 2021-12-09

**Authors:** Rohit Arora, Kenny M. Van Theemsche, Samuel Van Remoortel, Dirk J. Snyders, Alain J. Labro, Jean-Pierre Timmermans

**Affiliations:** 1Laboratory of Cell Biology and Histology, Department of Veterinary Sciences, University of Antwerp, 2610 Wilrijk, Belgium; rohit.arora@uantwerpen.be (R.A.); samuel.vanremoortel@uantwerpen.be (S.V.R.); 2Laboratory for Molecular, Cellular and Network Excitability, Department of Biomedical Sciences, University of Antwerp, 2610 Wilrijk, Belgium; kenny.vantheemsche@uantwerpen.be (K.M.V.T.); dirk.snyders@uantwerpen.be (D.J.S.); 3Department of Basic and Applied Medical Sciences, Ghent University, 9000 Ghent, Belgium

**Keywords:** GPCR, β-alanine, MRGPRD, constitutive receptor, Gq inhibitor, NF-kB, interleukin 6

## Abstract

G protein-coupled receptors (GPCRs) have emerged as key players in regulating (patho)physiological processes, including inflammation. Members of the Mas-related G protein coupled receptors (MRGPRs), a subfamily of GPCRs, are largely expressed by sensory neurons and known to modulate itch and pain. Several members of MRGPRs are also expressed in mast cells, macrophages, and in cardiovascular tissue, linking them to pseudo-allergic drug reactions and suggesting a pivotal role in the cardiovascular system. However, involvement of the human Mas-related G-protein coupled receptor D (MRGPRD) in the regulation of the inflammatory mediator interleukin 6 (IL-6) has not been demonstrated to date. By stimulating human MRGPRD-expressing HeLa cells with the agonist β-alanine, we observed a release of IL-6. β-alanine-induced signaling through MRGPRD was investigated further by probing downstream signaling effectors along the Gαq/Phospholipase C (PLC) pathway, which results in an IkB kinases (IKK)-mediated canonical activation of nuclear factor kappa-B (NF-κB) and stimulation of IL-6 release. This IL-6 release could be blocked by a Gαq inhibitor (YM-254890), an IKK complex inhibitor (IKK-16), and partly by a PLC inhibitor (U-73122). Additionally, we investigated the constitutive (ligand-independent) and basal activity of MRGPRD and concluded that the observed basal activity of MRGPRD is dependent on the presence of fetal bovine serum (FBS) in the culture medium. Consequently, the dynamic range for IL-6 detection as an assay for β-alanine-mediated activation of MRGPRD is substantially increased by culturing the cells in FBS free medium before treatment. Overall, the observation that MRGPRD mediates the release of IL-6 in an in vitro system, hints at a role as an inflammatory mediator and supports the notion that IL-6 can be used as a marker for MRGPRD activation in an in vitro drug screening assay.

## 1. Introduction

The G protein-coupled receptors (GPCRs) are the largest family of membrane receptors that play a key role in cellular signaling, regulating physiological processes. Apart from the classical complement and innate immunity pathways, we now know that GPCRs significantly contribute to acute and systemic chronic inflammations [1]. In the early 2000s, a novel subfamily of rhodopsin-like GPCRs was discovered in rodents and humans, showing substantial sequence homology with the MAS oncogene receptor and therefore named Mas-related genes (*Mrgs*) [2,3]. Initially, these Mrg receptors were thought to be mainly expressed by nociceptive neurons, where they are involved in modulating itch and pain, and were consequently referred to as sensory neuron-specific receptors (SNSRs) [4,5,6]. Later, the terms Mrgs and SNSRs were replaced by Mas-related G protein-coupled receptors (MRGPRs) [7]. Recent studies demonstrated that MRGPRs are also involved in hypersensitivity [8,9,10] and reported the presence of MRGPRs in other cell and tissue types, such as mast cells, macrophages, and cardiovascular tissue [11,12,13,14,15,16,17]. Several peptides and a few small molecules have been proposed as ligands for MRGPRs [5,6,18,19]. However, many of these ligands activate multiple receptors and share significant physiological overlap with other class-A receptors, because of which the majority of MRGPRs are still classified as orphan receptors [7]. Therefore, the pharmacological characterization and insight into the physiological roles of the majority of MRGPRs remain elusive.

One member of the MRGPR family, MRGPRD, is predominantly expressed in isolectin B4- positive (IB4+), small non-myelinated sensory neurons in dorsal root ganglia (DRG) and trigeminal ganglia (TG) of animals [2,20]. The β-alanine-activated rat MrgprD inhibits KCNQ/M K^+^ currents in DRG neuronal cultures, which points to MRGPRD involvement in modulating neuronal excitability [21]. In addition, activation of the mouse MrgprD receptor by β-alanine also has a regulatory impact on the transient receptor potential cation channel-A1 (TRPA1) and induces histamine-independent neuropathic itch and pain [22]. Moreover, in a recent publication, 5-oxoeicosatetraenoic acid (5-oxoETE), a polyunsaturated fatty acid (PUFA) metabolite, has been shown to induce somatic and visceral hyperalgesia without inflammation via the MRGPRD pathway, in this way triggering noxious symptoms in constipated irritable bowel syndrome (IBS) patients [23]. Altogether, MRGPRD seems to be involved in regulating nociception in neurons [24].

Additionally, MRGPRD has also been linked to the renin-angiotensin system (RAS) cascade of the cardiovascular system. It was found that angiotensin II (Ang II), angiotensin (1–7), and a derivative, alamandine, can activate MRGPRD [25,26,27]. Angiotensin II (Ang II) induces upregulation of cAMP, triggers phosphorylation of p-38, and induces fibrosis of rat vascular smooth muscle cells [28]. Intriguingly, these effects are mitigated by alamandine, which attenuates Ang II-associated hypertension and cardiac remodeling. Alamandine-activated rat MrgprD also induces NO release, suggesting that MRGPRD could also be involved in cardioprotection, controlling vasodilation and fibrosis in the heart [29,30]. Genetic ablation of *MrgprD* in mice led to a reduced mechanical nociception ability of sensory neurons and caused dilated cardiomyopathy [17,31].

MRGPRD involvement is not limited to neurons and cardiovascular tissues; high expression levels of MRGPRD have also been reported in human lung cancer tissues, where it promotes cell proliferation and tumorigenicity [32]. Additionally, L-βAIBA, a structural analog of β-alanine and a secretory metabolite of muscle cells, promotes the survival of MRGPRD-expressing mouse osteocytes by maintaining mitochondrial integrity, hence improving bone formation [33]. 

Several lines of evidence now indicate that MRGPR-mediated signaling is linked to increased production and release of inflammatory cytokines. Cells expressing MRGPRX1, another member of the MRGPR family, have been shown to release the pro-inflammatory cytokine interleukin-6 (IL-6) upon activation of the receptor through cleavage of its N-terminus by cysteine protease Der p1 [34]. In another study, alamandine-activated mouse MrgprD reduced the levels of the pro-inflammatory cytokines tumor necrosis factor-alpha (TNF-α) and interleukin-1 beta (IL-1β) in lipopolysaccharide (LPS)-stimulated macrophages [15]. Furthermore, a recent study hinted at mouse MrgprD involvement in LPS-induced inflammatory pain and activation of the NF-kB signaling pathway mediated via IkB kinases (IKKα and IKKβ) [35]. Therefore, the MRGPRD-mediated activation of NF-κB as well as the release of inflammatory cytokines (TNF-α and IL-1β) largely suggest a much broader role of MRGPRs in inflammation and immune biology than assumed thus far. Given the lack of experimental proof that agonist-activated human MRGPRD could mediate the release of the pleiotropic cytokine IL-6, in the present study, the release of IL-6 from MRGPRD-expressing cells was analyzed.

## 2. Results

### 2.1. β-Alanine-Mediated Activation of Human MRGPRD Induces Release of IL-6

MRGPRD can be activated by β-alanine [36,37], a derivative of degraded carnosine. Carnosine (β-alanyl-L-histidine) is a dipeptide molecule that is largely stored in mammalian skeletal muscles and, to some extent, in brain neurons and heart muscle [38,39]. Carnosine acts as a physiological buffer and scavenges reactive oxygen species to maintain cell homeostasis [40]. Carnosinase breaks down carnosine into β-alanine and L-histidine and elevates the concentration of β-alanine in plasma. 

β-alanine-activated MRGPRD expressing cells predominantly upregulate inositol phosphates (IP3/IP1), presumably through the activation of the G protein α-subunit. ‘Gαq’ then activates phospholipase C (PLC), which hydrolyses phosphatidylinositol 4,5-biphosphate (PIP2) to generate diacylglycerol (DAG) and inositol triphosphate (IP3). In addition, up to a limited extent, the stimulation of MRGPRD with β-alanine also inhibits adenylyl cyclase activity and reduces cAMP production, which hence is most likely mediated by the G protein α-subunit ‘Gαi’ [21,36]. However, the role of the MRGPRD-activated ‘Gαq’ signaling cascade in regulating cytokine release has not been demonstrated. Here, we sought to determine whether β-alanine-mediated activation of human MRGPRD triggers the release of IL-6. To this end, MRGPRD expressing HeLa cells were treated with either β-alanine (100 µM) or vehicle control. A significant, nearly 7-fold higher release of IL-6 was observed from MRGPRD-expressing cells treated with β-alanine, compared to β-alanine-treated control cells (transfected with empty vector; pCMV-GS). In addition, IL-6 levels of non-treated MRGPRD-expressing cells were nearly 4-fold higher than those of non-treated control cells (Figure 1). The difference in IL-6 release between β-alanine-treated and non-treated MRGPRD-expressing cells was 1.6-fold, which was found to be non-significant. These findings are indicative of high basal activity of MRGPRD. 

Evidence from literature indicates that many cell types, although not all, are capable of producing IL-6 and therefore able to release it [41]. To evaluate if the IL-6 release was not limited to HeLa cells, we further tested whether β-alanine-mediated activation of MRGPRD expressed in another cell type, i.e., HT1080, could also induce IL-6 release. To this end, HT1080 cells expressing MRGPRD were treated with either β-alanine (100 µM) or vehicle control. IL-6 release was observed from MRGPRD-expressing HT1080 cells treated with β-alanine, compared to β-alanine-treated control cells (Appendix A).

### 2.2. Demonstration of Basal Activity of MRGPRD and Assay Optimization to Maximize the IL-6 Detection Window

Many GPCRs display elevated basal activity under physiological conditions, which poses challenges in terms of understanding the regulatory mechanisms of receptors and hinders drug discovery for therapeutic purposes [42,43,44]. Our data indicate that MRGPRD was basally active and induced IL-6 release (Figure 1).

To further elucidate the IL-6 release from non-treated MRGPRD-expressing cells (Figure 1), i.e., the basal activity of MRGPRD, we transiently transfected HeLa cells with gradually increasing amounts of plasmid cDNA per well. Our data revealed an MRGPRD expression-dependent release of IL-6 (Figure 2A), which confirms the high basal activity of MRGPRD. Further stimulation of the MRGPRD-expressing HeLa cells with the agonist β-alanine did not show a significant increase in IL-6 release above the basal levels. As a control, human protease-activated receptor 2 (PAR2) was utilized to demonstrate that, in contrast to MRGPRD-expressing cells, PAR2-expressing cells did not show a PAR2 expression-dependent release of IL-6 (Figure 2B). Since it is known that the PAR2-activating peptide SLIGKV-NH_2_ induces the release of IL-6 from PAR2-expressing cells [34], our experimental control consisted in treating PAR2-expressing cells with the agonist SLIGKV-NH_2_, which clearly shows the PAR2 expression-dependent release of IL-6 into the medium (Figure 2B).

Having observed that MRGPRD displays high basal activity and that the effect of β-alanine stimulation on MRGPRD-expressing cells under normal physiological conditions (i.e., in the presence of fetal bovine serum; FBS) was marginal as compared to non-treated MRGPRD, we first investigated whether this activity originated from the presence of FBS, which constituted 10% of the cell culture medium [36]. Traces of ligands in FBS may keep the receptors engaged in an active state of signaling. To address this, we modified the stimulation protocol and exchanged the cell culture medium for the medium without FBS after transfection. We found that MRGPRD-expression was impaired when cells were cultured immediately in the medium without FBS upon transfection. Therefore, in an adapted protocol, we allowed the cells to express MRGPRD for 24 h before switching to medium without FBS. The first adapted protocol consisted of stimulating the MRGPRD-expressing cells with β-alanine at the time of switching to medium without FBS, i.e., 24 h post-transfection, and collecting samples for IL-6 analysis 24 h later, i.e., 48 h post-transfection (Figure 3A). The fold change observed between vehicle-treated and β-alanine-treated MRGPRD-expressing cells was approximately 2-fold, which was significantly higher as compared to the protocol using a medium with FBS (shown in Figure 1). Since the IL-6 readout for MRGPRD activation is far downstream of the signaling cascade, the cells most likely need to be cultured for a longer period in a medium without FBS in order to reduce the high basal activity of MRGPRD. To test this, the protocol was adjusted a second time, and the cells were kept in medium without FBS (starvation) for 24 h prior to being stimulated with β-alanine. The medium was replaced by the medium without FBS at 24 h post-transfection, and the cells were cultured for another 24 h. Subsequently, the medium was refreshed by the medium without FBS, and β-alanine was added. Following this protocol, stimulation of the cells was initiated at 48 h post-transfection. After another 24 h of incubation, i.e., 72 h post-transfection, the samples were collected, and IL-6 analysis was performed (Figure 3B). Interestingly, the fold change observed between vehicle-treated and β-alanine-treated MRGPRD-expressing cells increased to approximately 5-fold, which was significantly higher and offered a good range of detection for IL-6 as an assay for activation of MRGPRD. Additionally, a comparison of the two protocols revealed that the release of IL-6 from non-treated MRGPRD-expressing cells was reduced by 2.4-fold when culturing the cells for a longer period in medium without FBS (Figure 3A vs. Figure 3B). These data indicate that the basal activity of MRGPRD can indeed be attributed to either so far unidentified (partial) ligands or substances linked to ligand-independent GPCR effectors present in FBS, which is why we decided to use the adapted protocol for further experiments.

### 2.3. FBS Is Associated with Activation of MRGPRD and Induces Release of IL-6

Having optimized the assay, we next investigated the agonistic effect of FBS on MRGPRD using the above-described protocol (Figure 3B). The HeLa cells expressing MRGPRD were treated with 10%, 3%, or 1% FBS-containing DMEM, and FBS concentration-dependent increase in IL-6 release was observed. The MRGPRD cells stimulated with 10% and 3% FBS displayed a significant, 2.4-fold increase in IL-6 release compared to control cells (pCMV-GS transfected cells) treated with similar percentages of FBS (Figure 4). Cells expressing MRGPRD and treated with 1% FBS, as expected, exhibited lower IL-6 levels as compared to cells cultured in 10% and 3% FBS medium. Compared to pCMV-GS transfected control cells cultured in a 1% FBS medium, the MRGPRD-expressing cells displayed a 2.1-fold higher IL-6 production. Moreover, the MRGPRD-expressing cells incubated without (0%) FBS released the lowest amount of IL-6, which was still 1.5-fold higher than that of its respective control cells. Together these findings indicate that, in addition to being activated by FBS, MRGPRD also displays constitutive activity, which is in agreement with a recent publication [44].

Having demonstrated that the high basal activity of MRGPRD is due to the presence of FBS (Figure 4), we next examined whether the presence or absence of FBS affects the potency of β-alanine and alters the concentration–effect curve. In all published reports of β-alanine concentration–effect curves and EC_50_ values, FBS-containing medium was used to culture cells, and/or the cells were starved in medium without FBS for only 1 h [27,37,45]. Therefore, we used our optimized timeline protocol described in Figure 3B to define the EC_50_ of β-alanine for IL-6 release. Stimulation of HeLa cells expressing MRGPRD with different concentrations of β-alanine in the presence (Figure 5A) or absence (Figure 5B) of FBS yielded an EC_50_ of 151 ± 14 µM (*n* = 3) and 125 ± 10 µM (*n* = 4), respectively. Furthermore, no IL-6 release was observed in cells expressing empty vector (pCMV-GS) when treated with β-alanine in the presence or absence of FBS. Although MRGPRD exhibited higher basal activity in FBS conditions compared to the FBS-free condition, the obtained EC_50_ values were comparable. However, a comparison of the dynamic range by limiting the output to 10% (EC_10_) and 90% (EC_90_) of the maximal response revealed a higher dynamic range for IL-6 detection in FBS-free conditions. The dynamic range detected for IL-6 as an assay for MRGPRD activation with FBS and in the FBS-free condition was 2.9 and 4.6, respectively.

### 2.4. Inhibiting MRGPRD-Gαq/PLC/IKK/NF-κB Signaling Impedes the Release of IL-6

β-alanine-activated MRGPRD predominantly couples to Gαq protein, which upon activation stimulates PLC. Activated PLC further catalyzes the hydrolysis of PIP2 into IP3 and DAG. IP3 is subsequently dephosphorylated to generate IP1 (Appendix A) [37]. In several studies, it has been demonstrated that inhibition of PLC is able to block the activation of the NF-kB signaling cascade, which regulates inflammatory gene transcriptions. In order to validate this assumption in our experimental setting, we created a HeLa cell line stably expressing MRGPRD. We first investigated whether β-alanine-mediated activation of MRGPRD could induce NF-κB phosphorylation in HeLa cells. Stimulation of the MRGPRD-expressing HeLa cells with β-alanine increased NF-κB phosphorylation, whereas no upregulation of NF-kB phosphorylation was observed in control HeLa cells (Figure 6A). Furthermore, MRGPRD-expressing HeLa cells treated with a Gαq protein inhibitor (YM-254890), a pan-PLC inhibitor (U-73122) or an IKK complex inhibitor (IKK-16) [46,47,48,49,50] for 1 h prior to stimulation with β-alanine, blocked the phosphorylation of NF-kB as compared to vehicle-treated MRGPRD-expressing cells (Figure 6A).

Likewise, we examined whether the IL-6 release resulting from the β-alanine-activated MRGPRD-Gαq/PLC/NF-κB canonical axis could be prevented by the Gαq inhibitor, the pan-PLC inhibitor, or the IKK complex inhibitor. To this end, HeLa cells stably expressing MRGPRD were treated either with vehicle or with the respective inhibitors for 1 h prior to stimulation with β-alanine. The MRGPRD-expressing cells pre-incubated with the Gαq protein inhibitor or the IKK complex inhibitor revealed a significant decrease in IL-6 release, whereas pre-incubation with the PLC-inhibitor showed a ~40% reduction in IL-6 release (Figure 6B).

## 3. Discussion

GPCRs are capable of transducing extracellular signals into the regulation of downstream signaling pathways that control cellular responses and regulate gene expressions. Therefore, GPCRs have become a major target of therapeutic interventions [51]. Nevertheless, our knowledge of the GPCR-mediated inflammatory responses is still limited. In this study, we demonstrate that HeLa cells expressing MRGPRD induced the release of the pleiotropic cytokine IL-6 when activated by its agonist β-alanine (Figure 1). 

Considering that MRGPRD has been reported as constitutively active in physiological conditions [37], we tried to further define the constitutive (ligand-independent) (Figure 4) and basal activity of MRGPRD (Figure 2A). Our results showed that MRGPRD expressing HeLa cells in ligand-independent conditions (i.e., in the absence of agonist or FBS) release IL-6 and that IL-6 release in these cells was increased in normal physiological conditions (i.e., in the presence of FBS). The high basal activity could be attributed to the FBS, which may contain physiological concentrations of unrecognized MRGPRD-activating ligands such as β-alanine, β-aminoisobutyric acid (L-βAIBA), gamma-aminobutyric acid (GABA), etc. [37]. Assessment of the concentration–effect curves obtained in the medium with and without FBS did not reveal a significant difference between β-alanine EC_50_ values (Figure 5A,B), although the dynamic window for IL-6 detection increased in the medium without FBS. Taken advantage of the increased dynamic window for IL-6 detection, the assay could be utilized for screening allosteric modulators in the presence and absence of ligand for the receptor.

In HeLa cells, β-alanine preferentially activates the MRGPRD-Gαq signaling pathway (IP1 induction; Appendix A), which leads to IL-6 release (Figure 6B). Several studies have revealed that cytokine expression is dependent on nuclear translocation of phosphorylated NF-κB to regulate inflammatory gene transcriptions [52]. Accordingly, our results demonstrate the activation of NF-κB (Figure 6A), which was blocked by a Gαq protein inhibitor, a pan-PLC inhibitor, and an IKK complex inhibitor (IKKα and IKKβ inhibitor). 

The IKK complex is the key regulator of the NF-κB cascade, which consists of the IKKα and IKKβ kinases, as well as of the regulatory subunit IKKγ/NEMO. Activated IKK kinases phosphorylate inhibitory IkBα protein, which leads to ubiquitination and degradation of IkBα to unmask the activated NF-κB to translocate to the nucleus. Our results demonstrated that the IKK-16 inhibitor decreased the activation of NF-kB, which in turn reduced the IL-6 release from β-alanine-stimulated MRGPRD-expressing HeLa cells. Moreover, the observed attenuation of IL-6 release by the Gαq protein inhibitor and the intermittent inhibition of IL-6 release by the pan-PLC inhibitor (Figure 6B) hint at involvement of other signaling pathways that ultimately converge into the NF-κB cascade. A contribution of, for example, G-protein-coupled receptor kinases, β-arrestin recruitment, or IL-6 mediated feedback loops regulating IL-6 release from MRGPRD expressing HeLa cells cannot be excluded [53,54,55,56]. Altogether, our results unambiguously demonstrate the importance of MRGPRD-Gαq/PLC/NFκB signaling in the regulation of this pro-inflammatory cytokine IL-6.

Several recent studies as well as our data clearly show the regulatory role of the MRGPRD in inflammatory cytokine release. Hence, a more profound insight into the molecular mechanisms and scaffold proteins regulating the inflammation through MRGPRD might present an opportunity for further development of targeted therapeutic interventions.

## 4. Materials and Methods

### 4.1. Materials

DNA oligonucleotides were obtained from IDT (Leuven, Belgium). Plasmid preparations were performed using either the plasmid miniprep or maxiprep kit from Macherey-Nagel (Düren, Germany). The PCR/gel clean-up kit used to purify PCR-amplified products was from Macherey-Nagel. Agar, LB broth (high salt), SOC medium, ampicillin, and kanamycin were purchased from Sigma-Aldrich (Merck; Kenilworth, NJ, USA). Restriction enzymes and T4 DNA ligase were acquired from New England Biolabs (Ipswich, MA, USA). XL2-Blue ultracompetent cells used for transformation were purchased from Stratagene (San Diego, CA, USA). Pfu DNA polymerase was obtained from Promega (Madison, WI, USA). Human MRGPRD cDNA (pENTR223.1-MRGPRD; Cat# HsCD00080297) was obtained from the DNASU [57] plasmid repository (Tempe, AZ, USA), plasmid pcDNA3.1(+)-PAR2 was a generous gift from Dr. Rithwik Ramachandran, plasmid p-NCS-Antares (Cat# 74279) coding NanoLuc (NLuc) was obtained from Addgene (Cambridge, MA, USA), and pCMV-ECFP-N1 was from Clontech (Palo Alto, CA, USA). Dulbecco’s Modified Eagle Medium (DMEM) (Cat# 41966029), fetal bovine serum (FBS) (Cat# 10270-106), penicillin-streptomycin (Cat# 15140122), and Dulbecco’s phosphate-buffered saline (DPBS; Cat# 14190169) were purchased from Gibco (Waltham, MA, USA); 100 mm cell culture dishes (CELLSTAR; Cat# 664160), 6 well cell culture plates (CELLSTAR; Cat# 657160), and 96 well black well plate (CELLSTAR; Cat# 655090) were purchased from Greiner Bio-One (Frickenhausen, Germany). β-alanine (Cat# 05160) was purchased from Sigma-Aldrich. Human PAR2 agonist (SLIGKV-NH_2_; Cat# 3010) was obtained from Tocris (Abingdon, UK). The PLC inhibitor (U 73122; Cat# 1268) was purchased from Tocris, the Gq inhibitor YM 254890 (Cat# 10-1590-0100) from Focus Biomolecules (Plymouth, PA, USA), and the IKK complex inhibitor (IKK-16; Cat# S2882) from Selleck Chemicals GmbH (Berlin, Germany). Quantikine ELISA human IL-6 (Cat# D6050) was purchased from R&D Systems (Minneapolis, MN, USA). IP-One-Gq HTRF (Cat# 62IPAPEB), phospho-NF-κB (Ser536) cellular HTRF (Cat# 64NFBPET), Total NF-κB cellular HTRF kits (Cat# 64NFTPEG), and HTRF 96 well low volume white plate (Cat# 66PL96025) were acquired from Cisbio (Codolet, France). The Bradford assay kit (Cat# 23246), pierce IP lysis buffer (Cat# 87787), halt protease inhibitor cocktail (Cat# 87786), Geneticin (G418 Sulphate; Cat# 10131027), and Lipofectamine 2000 (Cat# 11668019) were obtained from Thermo Fisher Scientific (Waltham, MA, USA). The antibodies used in this study were anti-HA rat high-affinity IgG (Roche, Cat# 11867432; Sigma-Aldrich), anti-rat IgG conjugated with horseradish peroxidase (HRP) (Cat #9037; Sigma-Aldrich), and a β-actin rabbit monoclonal antibody (Cat# 4970S; Cell Signaling Technology), anti-Rabbit IgG conjugated to HRP (Cat# NBP1-75283; Novus Biologicals). NuPAGE novex 4-12% Bis-Tris gels (Cat# NP0321), NuPAGE MOPS running buffer (Cat# NP001), NuPAGE transfer buffer (Cat# NP00061), NuPAGE LDS sample loading buffer (Cat# NP007), Restore Plus Western blot stripping buffer (Cat# 46430), Pierce ECL Plus Western blotting substrate (Cat# 32132) and BenchMark pre-stained protein ladder (Cat# 10748010) were purchased from Thermo Fisher Scientific. Amersham Hybond-P-polyvinylidene difluoride (PVDF) membrane (Cat# RPN303F) was obtained from GE Healthcare (Boston, MA, USA).

### 4.2. Plasmid Preparation

Human MRGPRD cDNA was PCR-amplified from pENTR223.1-MRGPRD using Pfu DNA polymerase. The amplified PCR fragment was digested with the restriction enzymes HindIII/BamHI and ligated by a T4 DNA ligase in pCMV-ECFP-N1 to generate the pCMV-MRGPRD-ECFP-N1 plasmid. A stop codon was introduced at the C-terminus of MRGPRD by site-directed mutagenesis to generate the pCMV-hMRGPRD plasmid. Subsequently, MRGPRD was genetically tagged at its C-terminus with a human influenza hemagglutinin (HA; YPYDVPDYA) just before the stop codon to generate the pCMV-MRGPRD-HA plasmid. Similarly, human Protease-activated receptor 2 (PAR2) cDNA was PCR-amplified from pcDNA3.1(+)-PAR2 and inserted in-frame in pCMV-MRGPRD-HA (replacing MRGPRD) to generate pCMV-PAR2-HA. A blank/empty vector was generated by first introducing an XhoI restriction enzyme (RE) recognition site into pCMV-EYFP-N1 at the C-terminal end of EYFP to generate pCMV-EYFP-Xho1-N1. Subsequently, the pCMV-EYFP-XhoI-N1 plasmid was digested with NheI/XhoI, and an annealed oligonucleotide was inserted to generate the pCMV-GS (pCMV-MCS-GS-MCS-GS-MCS) plasmid, having multiple cloning sites connected by a flexible linker of glycine and serine residues (2xGGGGS). The NLuc gene sequence without start codon was PCR amplified from p-NCS-Antares and inserted in-frame between BamHI/XhoI to generate pCMV-GS-NLuc plasmid. The human MRGPRD cDNA was PCR-amplified from the pCMV-MRGPRD-HA plasmid and inserted in-frame between EcoRI/SalI sites to generate the pCMV-MRGPRD-NLuc plasmid. All plasmids were verified by Sanger sequencing at the VIB Genomic core (VIB-Centre for Molecular Neurology, University of Antwerp).

### 4.3. Cell Culture, Transfections, Treatments, and IL-6 Detection

HeLa cells were cultured in 100 mm dishes in a humidified incubator at 37 °C with 5% CO_2_ using DMEM supplemented with 10% FBS and 1% penicillin-streptomycin. For the ELISA experiments, HeLa cells were collected by trypsinization, and 2.5 × 10^5^ cells/well were seeded in a 6-well cell culture plate. Sixteen hours post-seeding, the medium was removed, and cells were washed once with 1 ml phosphate-buffered saline (PBS). Cells were transfected with 2 µg plasmid cDNA using Lipofectamine 2000 as per the manufacturer’s instructions. After 4 h, the transfection medium was replaced with 2 ml DMEM supplemented with 10% FBS and 1% penicillin-streptomycin, and the plates were returned to an incubator. Whenever necessary to obtain 2 µg of plasmid/well for transfection, the remaining amount was compensated with the pCMV-GS (empty) plasmid during transfection. 

Twenty-four hours post-transfection, the cell supernatant medium was removed. At every medium removal, cells were washed once with 1 ml PBS. The experiment timeline and treatment conditions are shown in each figure where appropriate. Two milliliters of DMEM with or without 10% FBS was added to each well containing the desired concentration of β-alanine, SLIGKV-NH_2_, or FBS. The plates were returned to the incubator for another 24 h before sample collection.

After the desired incubation, cell supernatant medium was collected in a 2 mL microcentrifuge tube and spun at 1000× *g* for 5 min at 4 °C to pellet down debris. Without disturbing the pellet, the supernatant medium was transferred into a fresh microcentrifuge tube and stored at −80°C until assayed by ELISA. For IL-6 assessment, the frozen supernatant medium was thawed on ice, samples were diluted (1:200) in assay diluent buffer, and IL-6 estimation was performed using an IL-6 ELISA kit as per the manufacturer’s instructions. For protein sample collection, cells were first washed with 1 mL of PBS followed by the addition of 300–500 µL IP-lysis buffer (supplemented with 1% protease inhibitor) to each well. Subsequently, the plates were maintained on ice for 20 min with intermittent shaking every 5 min. Lysate was collected in a microcentrifuge tube, spun at 13,000× *g* for 10 min at 4 °C; the resulting protein supernatant was transferred into a new microcentrifuge tube and stored at −80 °C for further analysis with a Bradford assay, as per the vendor’s instructions. The calculated IL-6 (ng/mL) concentrations from the collected supernatant medium were normalized to the protein concentrations (mg/mL) from the respective well. IL-6 release is represented in ng/mg.

### 4.4. Immunoblotting

The protein samples collected as described in Section 4.3 were utilized for immunoblotting. Protein estimation was performed using Bradford reagent. Samples were prepared in IP-lysis buffer, supplemented with 5 µL of 4× NuPAGE LDS sample loading buffer, and heated at 95 °C for 3 min. Protein samples (20 µg) were resolved on a NuPAGE 4–12% Bis-Tris gel using NuPAGE MOPS SDS running buffer (200 V for 50 min). Proteins were transferred onto the PVDF membrane by wet blotting using 20% methanol 1× NuPAGE transfer buffer (100V for 1 h). The blots were incubated overnight in blocking buffer (5% skimmed milk dissolved in PBS with 0.1% Tween 20) at 4 °C and subsequently probed with anti-HA rat IgG (1:1000 in blocking buffer) for 1 h at room temperature (RT), followed by incubation with anti-rat IgG conjugated with HRP (1:2000 in blocking buffer) for 1 h at RT. Chemiluminescence signals were detected using ECL plus substrate in an Amersham Imager 600. Subsequently, the blots were stripped using 5 mL restore plus Western blot stripping buffer, washed once with PBS, and incubated overnight in blocking buffer at 4 °C. On the next day, the blots were probed with anti-β-actin rabbit monoclonal antibody (1:2000 in blocking buffer) for 1 h at RT, followed by incubation with anti-rabbit IgG conjugated to HRP (1:10,000 in blocking buffer) for 1 h at RT and the chemiluminescence signals were recorded again using ECL plus substrate in an Amersham Imager 600.

### 4.5. MRGPRD Stable Cell Preparation and IL-6 Inhibition

MRGPRD-NLuc stable cells were developed by transfecting the pCMV-MRGPRD-NLuc plasmid in HeLa cells, followed by antibiotic selection of stable clones using Geneticin (1000 µg/mL). 

HeLa (mock) and HeLa cells stably expressing MRGPRD-NLuc were trypsinized, and 2.5 × 10^5^ cells/well were seeded in a 6-well cell culture plate. At 16 h post-seeding, the medium was removed, and the cells were washed once with 1 mL PBS. Subsequently, the medium was replaced by 2 mL DMEM (overnight starvation), and the plates were returned to the incubator (37 °C with 5% CO_2_). After 24 h of serum starvation, the medium was replaced again by either 2ml DMEM containing vehicle control (percentage of DMSO was 0.1%) or 2 ml DMEM containing the desired concentration of inhibitor (inhibitor dissolved in DMSO; 0.1%). The plates were kept in the incubator for 1 h. Afterward, the cells were stimulated with vehicle (percentage of milli-Q was 0.2%) or β-alanine (dissolved in milli-Q; 0.2%). The final concentration of β-alanine was 100 µM. The plates were then returned to the incubator for another 8 h before sample collection (supernatant and protein collection, as described in Section 4.3). The collected supernatant samples were thawed on ice and diluted (1:5), and IL-6 estimation was performed using the IL-6 kit as per the manufacturer’s instructions. Protein estimation was done using Bradford reagent. The IL-6 concentration was normalized to the protein concentration, expressed as ng/mg. The relative fold change (ΔF) was derived by normalizing IL-6 (ng/mg) to that of its respective controls, i.e., vehicle-treated (DMSO) or inhibitor-treated (inhibitor dissolved in DMSO, without β-alanine). The fold-change (ΔF) was further normalized to the ΔF of β-alanine-stimulated MRGPRD-expressing cells (i.e., ΔFmax) and expressed as a percentage. 

### 4.6. NF-kB Detection and Inhibition Assay

HeLa (mock) and HeLa cells stably expressing MRGPRD-NLuc were trypsinized, and 3 × 10^4^ cells/well were seeded in 96 black well plates. At 24 h post seeding, the medium was removed and replaced by 100 µL DMEM (overnight serum starvation). On the next day, i.e., 48 h from seeding, the medium was replaced again by either with 50 µL DMEM containing vehicle control (percentage of DMSO was 0.1%) or 50 µL DMEM containing the desired concentration of inhibitor (dissolved in DMSO; 0.1%) for 1 h before stimulation with 50 µL DMEM containing vehicle control (DMSO; 0.1%) or inhibitor (dissolved in DMSO; 0.1%) with vehicle (percentage of milli-Q was 0.1%) or β-alanine (dissolved in milli-Q; 0.1%). The final concentration of β-alanine was 1 mM/well. Plates were then returned to the incubator (37 °C with 5% CO_2_) for another 10 min. After the desired treatment, the medium was removed, and 50 µL of lysis buffer supplemented with blocking buffer was added to the wells. The plates were kept on a shaker at 100 rpm for 30 min. Then, 16 µL cell lysate was transferred to an HTRF 96-well low volume white plate (Cisbio) for phospho-NF-κB (Ser 536) and total NF-κB estimation using Cisbio HTRF kits. Briefly, 4 µL of pre-mixed donor and acceptor antibodies were added to 16 µL of cell lysate and incubated overnight at 25 °C. On the next day, plates were read using an EnVision multimode plate reader. The calculated phospho-NF-κB HTRF ratios were normalized to total NF-κB HTRF ratios. The relative fold change (ΔF) was obtained by normalizing phospho- NF-κB/total NF-κB to its respective control, i.e., vehicle-treated (DMSO) or inhibitor-treated. The fold-change (ΔF) was further normalized to the ΔF of β-alanine-stimulated MRGPRD-expressing cells (i.e., ΔFmax) and expressed as percentages.

### 4.7. Data Analysis

Data analysis was performed using Microsoft Excel and GraphPad Prism 6 (San Diego, CA, USA). The concentration–effect curve and graphs were generated using GraphPad Prism 6. The *EC*_50_ values were obtained by fitting the concentration–effect curve with a Hill function. To determine the dynamic range, the limit was set to *EC*_10_ and *EC*_90_. Hereto, the *EC*_10_ and *EC*_90_ were calculated using Equation (1), where *F* was set to either 10 or 90, respectively.
(1)$$LogEC_{50}=LogECF-\left(\frac{1}{Hillslope}\right)*Log\left(\frac{F}{100-F}\right)$$

## Figures and Tables

**Figure 1 ijms-22-13254-f001:**
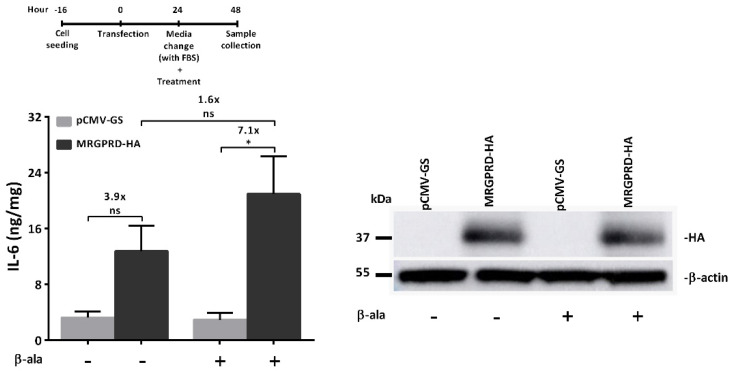
Activation of MRGPRD mediates IL-6 release. HeLa cells transiently expressing empty vector (pCMV-GS) or MRGPRD-HA were stimulated with either vehicle control or β-alanine (100 µM; final concentration/well). The normalized IL-6 release is represented in ng/mg. β-alanine- or vehicle-treated cells expressing MRGPRD released high and intermittent levels of IL-6, respectively. Whereas vehicle- or β-alanine-treated cells expressing the empty vector did not release significant amounts of IL-6. On the right, a representative Western blot of whole cell lysates from HeLa cells transiently transfected with either empty vector (pCMV-GS) or MRGPRD-HA plasmids is displayed. Expression of the MRGPRD-HA receptor (~37 kDa) was observed in both vehicle and β-alanine (100 µM)-treated MRGPRD-transfected cells, whereas cells transfected with the empty vector showed no expression (i.e., absence of a protein band of ~37 kDa). The graph represents the mean ± s.e.m values from three independent experiments. Statistical significance was determined using one-way analysis of variance (ANOVA), and Sidak’s post hoc test was applied for multiple comparisons. * *p* ≤ 0.05 was considered significant and ‘ns’ is non-significant.

**Figure 2 ijms-22-13254-f002:**
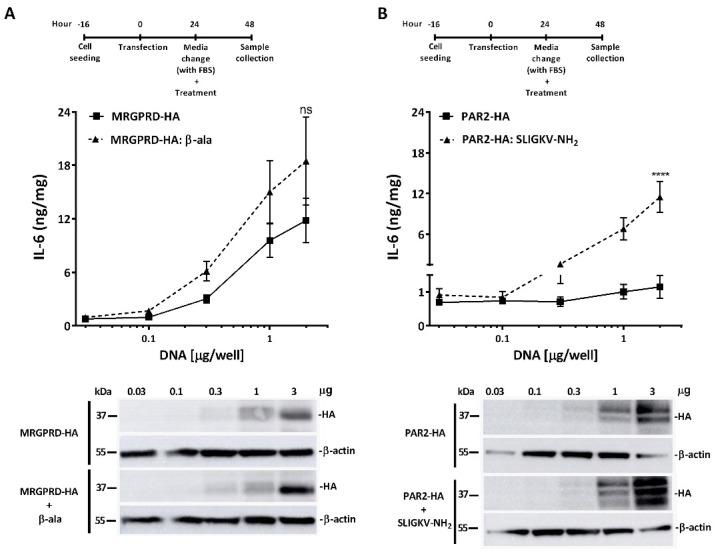
Basal activity of MRGPRD. HeLa cells expressing increasing concentrations of MRGPRD-HA and PAR2-HA receptors (transfected with 0.03, 0.1, 0.3, 1, and 2 µg of cDNAs per well). When treated with vehicle control, the MRGPRD-transfected cells did show an MRGPRD expression-dependent release of IL-6 (**A**)**,** while the PAR2-transfected cells did not show PAR2 expression-dependent release of IL-6 (**B**). Furthermore, when cells expressing increasing concentrations of MRGPRD-HA or PAR2-HA were treated with their respective agonists, β-alanine (100 µM) or the PAR2-agonist SLIGKV-NH_2_ (100 µM), a receptor concentration-dependent gradual IL-6 release was noticed (**A**,**B**). The release of IL-6 from vehicle-treated MRGPRD-HA cells points to basal activity of the receptor. Below the graphs, representative Western blot analysis of whole cell lysates from stimulated and non-stimulated HeLa cells transiently expressing increasing concentrations of MRGPRD-HA and PAR2-HA are shown. Each graph represents the mean ± s.e.m values from three independent experiments. The comparison between the two groups was analyzed by one-way ANOVA with Sidak’s post hoc test. **** *p* ≤ 0.0001 was considered significant and ‘ns’ is non-significant.

**Figure 3 ijms-22-13254-f003:**
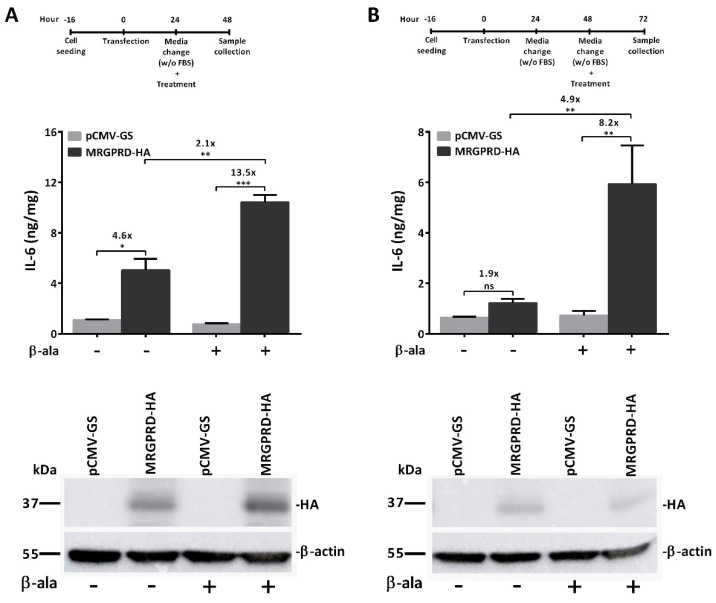
IL-6 assay optimization. To achieve a high dynamic window for the IL-6 detection, the experiment timeline was optimized. (**A**) At 24 h post-transfection, medium was replaced by FBS-free medium, and cells were treated with vehicle or β-alanine (100 µM). At 24 h post-treatment, i.e., at 48 h post-transfection, samples were collected for IL-6 detection. IL-6 levels from β-alanine-treated MRGPRD-expressing cells were 2-fold higher when compared to those from vehicle-treated MRGPRD-expressing cells. (**B**) Similarly, at 24 h post-transfection, medium was replaced by FBS-free medium for another 24 h, and at 48 h, the medium was once again replaced by FBS-free medium, and cells were treated with vehicle or β-alanine (100 µM). Samples were collected for IL-6 detection after 24 h of treatment, i.e., at 72 h post-transfection. IL-6 levels from β-alanine-treated MRGPRD-expressing cells were ~5-fold higher when compared to those from only vehicle-treated MRGPRD-expressing cells, which is substantially higher than for the protocol used in (**A**). The cells expressing empty vector (pCMV-GS), treated with β-alanine or vehicle, did not show IL-6 release above the basal level (**A**,**B**). Below the graphs are representative Western blots of cell lysates from HeLa cells transiently transfected with empty vector (pCMV-GS) or MRGPRD-HA plasmid, showing receptor expression in vehicle or β-alanine (100 µM)-treated MRGPRD-expressing cells subjected to the experiment timeline protocols illustrated in (**A**,**B**), respectively. Each graph represents the mean ± s.e.m values from three independent experiments. Statistical significance was determined using one-way ANOVA, and Sidak’s post hoc test was applied for multiple comparisons. * *p* ≤ 0.05, ** *p* ≤ 0.01, *** *p* ≤ 0.001 were considered significant and ‘ns’ is non-significant.

**Figure 4 ijms-22-13254-f004:**
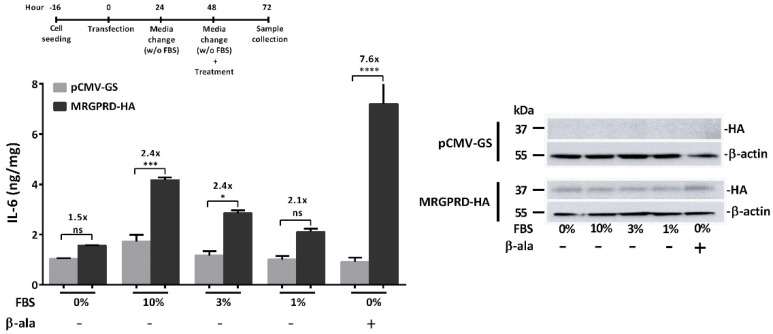
Agonistic effect of FBS and constitutive activity of hMRGPRD. HeLa cells expressing MRGPRD-HA or empty vector (pCMV-GS) were challenged with medium containing 10%, 3 %, 1% FBS and FBS-free (0%) medium. Significantly increased IL-6 levels were observed with increasing concentration of FBS from MRGPRD-expressing cells as compared to control cells (empty vector). β-alanine (100 µM)-treated MRGPRD-expressing cells served as positive control for this experiment. β-alanine- or vehicle-treated cells expressing the empty vector (pCMV-GS) did not show IL-6 release above the basal level. On the right, representative western blot of whole cell lysates from HeLa cells transiently transfected with empty vector or MRGPRD-HA plasmids, treated with vehicle or β-alanine (100 µM) under FBS (10%, 3%, and 1%) or FBS-free (0%) conditions. The graph represents the mean ± s.e.m values from three independent experiments. The comparison between the two groups was analyzed by one-way ANOVA with Sidak’s post hoc test. * *p* ≤ 0.05, *** *p* ≤ 0.001, **** *p* ≤ 0.0001 were considered significant and ‘ns’ is non-significant.

**Figure 5 ijms-22-13254-f005:**
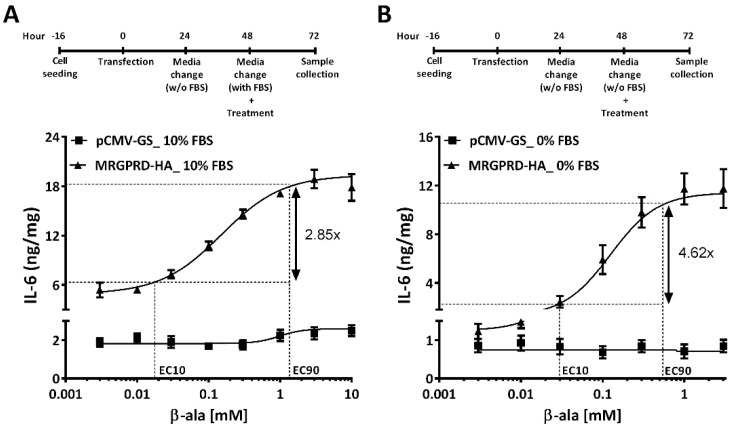
Concentration–effect curves of β-alanine obtained in the presence or absence of FBS. HeLa cells expressing the empty vector (pCMV-GS; square symbol) and MRGPRD-HA (triangle symbol) were treated with increasing concentrations of β-alanine in the absence or presence of FBS. Cells expressing MRGPRD induced the release of IL-6 in a concentration-dependent manner when treated with β-alanine with FBS (**A**) or without FBS (**B**). A higher basal activity of MRGPRD was observed in cells treated under FBS conditions as compared to non-FBS conditions. No significant IL-6 release was observed from cells expressing the empty vector (pCMV-GS). The dynamic range obtained when cells were treated in FBS (**A**) conditions was 2.85, whereas it increased to 4.62 when cells were treated in FBS-free conditions (**B**). Each graph represents the mean ± s.e.m values from at least three independent experiments.

**Figure 6 ijms-22-13254-f006:**
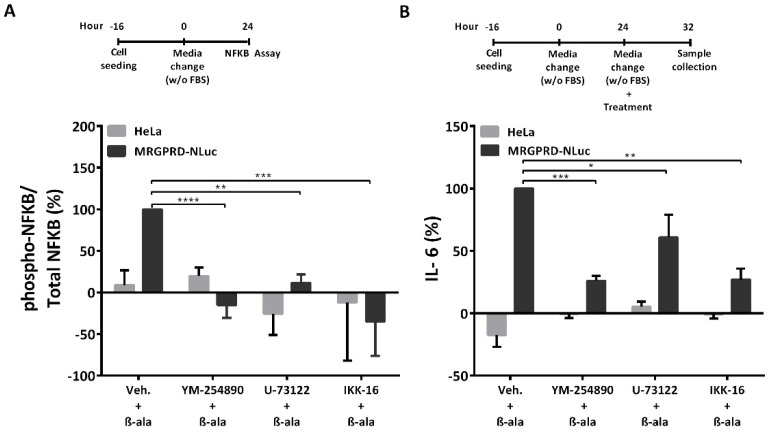
β-alanine-stimulated MRGPRD-induced IL-6 release is dependent on NF-kB activation. (**A**) HeLa cells stably expressing MRGPRD-NLuc were pre-treated with either vehicle control or the Gαq inhibitor (YM-254890; 10 µM), the pan-PLC inhibitor (U-73122; 10 µM) or the IKK inhibitor (IKK-16; 5 µM) for 1 h before being stimulated with β-alanine (1mM). β-alanine-stimulated MRGPRD-expressing HeLa cells induced NF-κB phosphorylation, whereas no induction was observed in cells pre-treated with either of the three applied inhibitors. (**B**) Similarly, HeLa cells stably expressing MRGPRD-NLuc were pre-treated with either vehicle control or Gαq (YM-254890; 10 µM) or pan-PLC (U-73122; 10 µM) or IKK (IKK-16; 5 µM) inhibitors for 1 h before being subjected to β-alanine (100 µM). MRGPRD-expressing cells upon stimulation with β-alanine showed a strongly reduced IL-6 release after being pretreated with either YM-254890 or IKK-16 and a less prominent but still significantly reduced IL-6 release compared to β-alanine stimulated MRGPRD expressing cells that were not pre-exposed with the inhibitor. Each graph represents the mean ± s.e.m values from three independent experiments. The comparison between the two groups was analyzed by one-way ANOVA with Sidak’s post hoc test. * *p* ≤ 0.05, ** *p* ≤ 0.01, *** *p* ≤ 0.001, **** *p* ≤ 0.0001 were considered significant and ‘ns’ is non-significant.

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
