# Peer review of "Constitutive, Basal, and β-Alanine-Mediated Activation of the Human Mas-Related G Protein-Coupled Receptor D Induces Release of the Inflammatory Cytokine IL-6 and Is Dependent on NF-κB Signaling"

_ijms, 2021, doi:10.3390/ijms222413254_

Round 1

Reviewer 1 Report

The present manuscript studies the release of pro-inflammatory cytokine IL-6 induced by human Mas-related GPCR D in the presence of b-alanine. The results presented in this manuscript are convincing related to the ability of b-alanine to induce section of IL-6 via MRGPRD. However, some experiments should be carried out to improve this manuscript:

-As shown by the authors b-alanine induce a dose-dependent activation of IL-6 secretion. To confirm the specificity of this action the use of antagonists such as MU-6840 could be demonstrated the specific action of b-alanine.

- The authors demonstrated that the presence of the YM-254890 Gq inhibitor reverse the b-alanine effect. However, to demonstrate that this effect is dependent of PLC (Phospholipase C) the use of the pan-inhibitor of this enzyme should be performed.

- The secretion of IL-6 is dependent of the NFkB activation. What is the impact of b-alanine on the activation of NFkB in the presented cell model? What is the impact of NFkB inhibitors? What is the impact of Gq and PLC inhibitors to this activation ? These aspects should be performed in the present study.

Reviewer 2 Report

In mammalian cells the largest family of membrane receptors is represented by the G-Protein Coupled Receptors (GPCRs), which are playing a key role in cellular signaling.

In the manuscript entitled "Constitutive, Basal and β-Alanine-Mediated Activation of Human Mas-Related G Protein-Coupled Receptor D Induces Release of Inflammatory Cytokine IL-6" Arora R. and colleagues report the release of the pro-inflammatory IL-6, in HeLa cells, as a readout of MRGPRD activation . After having assessed the optimal conditions to properly stimulate the MRGPRD with 2 different ligands, namely B-alanine and Alamandine, they found that only upon B-alanine stimulation the HeLa cells release IL-6. Eventually, by taking advantage of a pharmacological approach, using YM-254890, they noticed that inhibiting the Gαq impedes the IL-6 release.

To some extent the manuscript appears rather reductive and there are some major conceptual flaws. Hence, before publication quite few gaps must be fulfilled.

Major concerns:

  1. Figures 1 and 2 show the preliminary results that helped the authors to set-up proper protocol to stimulate cells with B-alanine and Alamandine. Serum contains dozens of molecules that can either cross-activate the receptor, thus giving background, or alternatively they may prevent or hinder the ligand binding. Thereby, in either cases distorting the results. Hence, I am wondering why authors stimulate cells with a ligand in presence of serum. It is pretty uncommon look at a receptor activation upon ligand stimulation in presence of serum. Thus, Figures 1 and 2 should be removed.
  2. It would be nice instead of working with transient transfection have stable clones. By working with transient transfection we are aware that transfection efficacy might be different from time to time. Therefore, it would be appreciated if, beside the plots (e.g. Fig. 2, 3 and 4), the authors could display a representative Western Blot showing the MRGPRD protein amount. In the case of Fig. 2 the MRGPRD protein amount should increase in a DNA dose dependent way. 
  3. Concerning the Fig. 7 please provide an internal control by assessing the IP1 accumulation as it has been done in Fig. S1.
  4. The Discussion section looks pretty scant. The authors should articulate little bit more this section. Many questions remain unanswered. Does the signal trigger Il-6 gene transcription? Is the IL-6 protein already present within the cells and thus just simply released? 
  5. The conclusion drawn by the authors "... our data support the hypothesis that IL-6 396 can be used as a biomarker for MRGPRD activation in an in vitro drug screening assay, and suggest that MRGPRD activation might contribute to inflammation and immune regulation in vivo..." it is not yet fully supported from their research findings. To draw such kind of conclusion one would need to extend these findings to, at least, a couple of more cell lines.
  6. Fig. 4 Please provide a detailed protocol how long was the incubation time for each condition. Moreover, from these data it seems that serum itself contains something that, when compared to the B-alanine alone, hinder the IL-6 release. This point should be also discussed and explained.

Minor concerns:

  1. typos (e.g. Sanger instead of sanger, line #335) 
  2. Within the section Materials & Methods for most of the reagents the authors provided also the catalog ID #, however many reagents lack it.
  3. Quite often in the legends to figure you indicate n=3 but then the authors stated that "...experiments were performed either in duplicate or triplicate....". Please, clarify this issue and provide the raw data.
  4. In the Figure 2B and Figure 5A & Figure 5B I noticed that there are values w/o error bars. Please draw the error bars. 

Round 2

Reviewer 1 Report

Authors have totally replied to my comments

Author Response

We like to thank the reviewer once again for the positive comments which have clearly improved our manuscript.

Reviewer 2 Report

I truly thank the authors for the efforts they made. I appreciated them very much. However, even in the revised version sometimes it happens that the authors draw sharp conclusion without having sufficient data as in the case of line 388. Sometimes it is more advisable keep conclusion a little bit more...open. I mean: why should it be an "unknown ligand"? Why not a ligand independent GPCR activation? We are aware that sometimes GPCR activation occurs in a ligand independent way by tyrosine kinase/s. But they represent minor issues. Consistently, I consider the revised version significantly improved, when compared to the previous one.

Author Response

We thank the reviewer for the kind words and appreciation for our efforts in improving the manuscript. We rephrased the corresponding paragraph (former line 388) as suggested (now lines 396-397).